# Intra-annual variations of spectrally resolved gravity wave activity in the UMLT region

René Sedlak[1], Alexandra Zuhr[1,2,a], Carsten Schmidt[2], Sabine Wüst[2], Michael Bittner[1,2], Goderdzi G. Didebulidze[3], Colin Price[4]

[1]Institute of Physics, University of Augsburg, Augsburg, Germany
[2]German Remote Sensing Data Center, German Aerospace Center, Oberpfaffenhofen, Germany
[3]Georgian National Astrophysical Observatory, Ilia State University, Tbilisi, Georgia
[4]Porter School of the Environment and Earth Sciences, Tel Aviv University, Israel
[a]now at: Alfred Wegener Institute, Helmholtz Center for Polar and Marine Research, Bremerhaven, Germany

*Correspondence to*: René Sedlak (rene.sedlak@physik.uni-augsburg.de)

**Abstract.** The period range between 6 min and 480 min is known to represent the major part of the gravity wave spectrum driving mesospheric dynamics. We present a method using wavelet analysis to calculate gravity wave activity with a high period-resolution and apply it to temperature data acquired with the OH* airglow spectrometers GRIPS (GRound-based Infrared P-branch Spectrometer) within the framework of the NDMC (Network for the Detection of Mesospheric Change; https://ndmc.dlr.de). We analyse data measured at the NDMC sites Abastumani in Georgia (ABA, 41.75° N, 42.82° E), ALOMAR in Norway (ALR, 69.28° N, 16.01° E), Neumayer III in the Antarctic (NEU, 70.67° S, 8.27° W), Observatoire de Haute-Provence in France (OHP, 43.93° N, 5.71° E), Oberpfaffenhofen in Germany (OPN, 48.09° N, 11.28° E), Sonnblick in Austria (SBO, 47.05° N, 12.95° E), Tel Aviv in Israel (TAV, 32.11° N, 34.80° E), and the Environmental Research Station Schneefernerhaus on top of Mt. Zugspitze, Germany (UFS, 47.42° N, 10.98° E). All eight instruments are identical in construction and deliver consistent and comparable data sets.

For periods shorter than 60 min, gravity wave activity is found to be relatively low and hardly shows any seasonal variability on the time scale of months. We find a semi-annual cycle with maxima during winter and summer for gravity waves with periods longer than 60 min, which gradually develops into an annual cycle with a winter maximum for longer periods. The transition from a semi-annual pattern to a primarily annual pattern occurs around a gravity wave period of 200 min. Although there are indications of enhanced gravity wave sources above mountainous terrain, the overall pattern of gravity wave activity does not differ significantly for the abovementioned observation sites. Thus, large-scale mechanisms such as stratospheric wind filtering seem to dominate the evolution of mesospheric gravity wave activity.

# 1 Introduction

Gravity waves represent an important coupling mechanism between different atmospheric regions by transporting energy and momentum not only horizontally but also vertically. While they are often generated in the lower atmosphere, their influence can even reach up to the ionosphere (Laštovička, 2006). It is widely accepted that gravity waves significantly determine global circulation patterns, most prominently the mean residual meridional circulation (Holton, 1983; Garcia & Solomon, 1985). They interact with large-scale dynamical structures such as planetary waves and are able to accelerate or decelerate atmospheric jets by momentum deposition (Hines, 1960; Hodges, 1969; Lindzen, 1981).

In order to take adequate account of their influence on global atmospheric dynamics – especially in view of predicting the effects of climate change – detailed knowledge of gravity wave parameters such as the amount of transported energy is essential. Especially in the UMLT (upper mesosphere / lower thermosphere) – the hot-spot region for gravity wave drag release (Gardner et al., 2002) – in situ measurements are hardly possible. Nevertheless, a variety of remote sensing techniques measuring at different locations allow the derivation of the potential energy density (Rauthe et al., 2008; Wüst et al., 2016, 2017a) or gravity wave activity (Gavrilov et al., 2004; Hibbins et al., 2007; Beldon & Mitchell, 2009; Offermann et al., 2009; Hoffmann et al., 2010, please note that 'gravity wave activity' can be calculated from variations of either wind or temperature and that different measurement techniques may be sensitive to different ranges of wave parameters) for different parts of the gravity wave spectrum. However, the results do not always agree concerning the intra-annual variability. This could be due to the different geographical positions of the observations and/or to the different data reduction and analysis algorithms and/or to the differing spectral ranges to which the instruments are sensitive (observational filter). Wüst et al. (2016, 2017a) showed that the spectral range is of importance in this context. The authors applied identical data reduction and analysis algorithms to temperature time series of identical OH* spectrometers and found that the gravity wave potential energy density (GWPED) evolves differently during the year for periods longer and shorter than 60 min.

We have analysed gravity wave activity based on data of eight infrared spectrometers called GRIPS (GRound-based Infrared P-branch Spectrometer) that are identical in construction and operated at different locations worldwide within the Network for the Detection of Mesospheric Change (NDMC, https://ndmc.dlr.de). The brightest component of the nocturnal airglow is created by excited hydroxyl molecules (OH*) in the UMLT emitting radiation in the visible and in the near infrared wavelength spectrum (Meinel, 1950; Leinert et al., 1998). The peak emission height of the OH* layer is located at about 87 km height on average (Baker & Stair, 1988; von Savigny et al., 2004; Melo et al., 2000; Wüst et al., 2017b). As gravity waves are causing local temperature fluctuations, they are modulating the emission behaviour of the OH* molecules (see for example Svenson & Gardner, 1998). This makes gravity waves and other dynamical features visible in the images of infrared cameras (see e.g. Taylor, 1997; Pautet et al., 2014; Hannawald et al., 2016; Sedlak et al., 2016; Hannawald et al., 2019; Wüst et al., 2019) but also in the time series of OH* rotational temperatures (Hines & Tarasick, 1987; Simkhada et al., 2009; Reisin & Scheer, 2001; Offermann et al., 2009; Wachter et al., 2015; Wüst et al., 2016, 2017a, 2018; Silber et al., 2017).

These are found to be in good agreement with the kinetic mesospheric temperatures (see e.g. Bittner et al., 2010; Noll et al., 2016) and can be derived from the line intensities of the OH* airglow radiation, which are measured with spectrometers (Mulligan et al., 1995; Espy & Stegman, 2002; Espy et al., 2003; French & Burns, 2004; Schmidt et al., 2013, 2018).

5    In this work, we make use of wavelet analysis, which allows us to derive spectrally-resolved gravity wave activity from time series of OH* rotational temperatures. In this way, we address gravity waves with periods in a comparatively broad range of $6-480\,\text{min}$ separately. Such a climatology resolving the spectrum of gravity waves is not available yet. The scientific focus of this paper is to reveal the period-dependence of the intra-annual cycles of gravity wave activity in the UMLT.

In section 2, the data sets we used are presented together with a description of the quality criteria we have applied. In section 10    3, we introduce the analysis and show the intra-annual cycles of period-resolved gravity wave activity. They are discussed in section 4 and the main results are summarized in section 5.

## 2 Data bases

GRound-based Infrared P-branch Spectrometers (GRIPS) are sensitive to electro-magnetic radiation in the near infrared. The intensities of the $P_1(2)$, $P_1(3)$, and $P_1(4)$ rotational lines of the OH(3-1) vibrational transition at 1.524, 1.534 and 1.543 µm are used to derive the rotational temperatures of the OH* airglow layer. This is done by assuming local thermodynamic equilibrium in the mesopause region (Noll et al., 2019). Measurements are only possible during the night-time since the OH* emission would not be detectable above the solar background.

GRIPS instruments are equipped with a 512 pixels InGaAs-photodiode array, which has maximum sensitivity between 1.5 and 1.6 µm. The temporal resolution ranges from 5 to 15 s. Further technical details can be found in Schmidt et al. (2013).

Measurements were started in 2009, when GRIPS 6 was put into operation at the DLR site Oberpfaffenhofen (OPN, 48.09° N, 11.28° E), Germany. The number of GRIPS instruments has increased since then. Today, fourteen instruments (GRIPS 5 to 18) are operated within the Network for the Detection of Mesospheric Change (NDMC, https://ndmc.dlr.de) providing extensive coverage around the globe. In this work, temperature series of eight different GRIPS instruments are used. Besides the GRIPS 6 time series, we analysed data acquired by GRIPS 8 at the Environmental Research Station Schneefernerhaus (UFS, 47.42° N, 10.98° E) below the summit of Mt Zugspitze in Germany, by GRIPS 16 at the Sonnblick Observatory (SBO, 47.05° N, 12.95° E) in Austria and by GRIPS 12 at the Observatoire de Haute-Provence (OHP, 43.93° N, 5.71° E) in France. The measurements above the Alps and their foothills are compared to data from the Lesser Caucasus (GRIPS 5 at the Abastumani Astrophysical Observatory, ABA, 41.75° N, 42.82° E in Georgia) and the Mediterranean (GRIPS 10 at Tel Aviv, TAV, 32.11° N, 34.80° E in Israel). We also analyse data from polar regions. GRIPS 9 was deployed at the Arctic Lidar Observatory for Middle Atmosphere Research, ALOMAR (ALR, 69.28° N, 16.01° E), Norway during a measurement campaign from winter 2010/11 to spring 2014. Measurements with GRIPS 15 are performed at Neumayer-Station III (NEU, 70.67° S, 8.27° W) in the Antarctic since March 2013. The start and end dates of the times series analysed in this publication are shown in Table 1 for each observation site.

As the temporal resolution of the GRIPS instruments is much better than necessary for retrieving gravity waves, the data sets are averaged to one-minute means in order to reduce noise. Before calculating the gravity wave activity from these time series, the data need to satisfy several criteria concerning data quality, data gaps and length of the data series. The data quality criteria agree with the ones presented in Wüst et al. (2016, 2017a): only temperature values which have an uncertainty (calculated by Gaussian error propagation) of less than or equal to 4.5 K and have been measured during episodes of a solar zenith angle larger than 100° (in order to avoid artefacts due to enhanced solar radiation during sunset or sunrise) are considered. Nocturnal temperature series are further analysed, if they consist of at least 240 successive values (corresponding to a time period of four hours). If more than one of such episodes is available for one night, we used only the longest one for further analysis.

In contrast to Wüst et al. (2016), who used an iterative approach of sliding means to calculate potential energy density, a wavelet analysis will be used in this work. This analysis method requires an equidistant time series. However, bad weather

(clouds) frequently causes data gaps. They are extrapolated based on the ten preceding data points by the maximum entropy method (MEM; see e.g. Ulrych and Bishop, 1975) in its capacity as a linear prediction filter (Bittner et al., 2000, Höppner & Bittner, 2007), if the data gaps are not larger than six minutes.

Due to meteorological and geophysical reasons, the monthly data coverage varies (Figure 1). For the high-latitudinal stations (NEU, ALR), it suffers from midnight sun during polar summer. Also the Alpine stations (SBO, UFS) show minimal observations during summer, principally due to bad weather. The station at TAV exhibits a rather inhomogeneous data distribution due to stray light and technical issues (Wüst et al., 2017a).

**Table 1. Start and end dates of the analysed time series for the respective station. The same start dates as in Wüst et al. (2016, 2017a) have been chosen for each data set as far as the respective stations have been analysed therein.**

| Station | Instrument | Start of analysed time series | End of analysed time series | Total number of nights | Number of nights observed | Number of nights analysed |
|---|---|---|---|---|---|---|
| ABA (41.75° N, 42.82° E) | GRIPS 5 | 2012/10/15 | 2018/06/05 | 2001 | 1974 | 853 |
| ALR (69.28° N, 16.01° E) | GRIPS 9 | 2011/01/01 | 2014/04/08 | 1192 | 875 | 277 |
| NEU (70.67° S, 8.27° W) | GRIPS 15 | 2013/03/18 | 2018/06/01 | 1900 | 1402 | 394 |
| OHP (43.93° N, 5.71° E) | GRIPS 12 | 2012/06/28 | 2018/06/01 | 2164 | 2152 | 882 |
| OPN (48.09° N, 11.28° E) | GRIPS 6 | 2011/01/08 | 2018/06/05 | 2704 | 2622 | 708 |
| SBO (47.05° N, 12.95° E) | GRIPS 16 | 2015/08/05 | 2018/05/12 | 1011 | 1006 | 235 |
| TAV (32.11° N, 34.80° E) | GRIPS 10 | 2011/11/25 | 2016/01/26 | 1523 | 1478 | 249 |
| UFS (47.42° N, 10.98° E) | GRIPS 8 | 2011/01/05 | 2018/06/02 | 2704 | 2646 | 835 |

## 3 Analysis and results

The wavelet analysis is a time-dependent spectral analysis method. In contrast to other analyses, e.g. the harmonic analysis, which assumes stationary periodic signatures (Bittner et al., 1994), the wavelet analysis can identify transient wave signals, which makes it well suited for the identification of gravity wave signatures. A comprehensive mathematical description of the wavelet analysis can be found in Chui (1992). We use the wavelet analysis as it was described by Ochadlick et al. (1993) based on a Morlet wavelet and apply it to the temperature time series of each night. The wavelet analysis method then provides a two-dimensional wave spectrum that depends on time and period (sampling rate of 1 min in both domains).

In order to perform a significance test, the wavelet analysis is repeated another eleven times for randomly generated data (white noise), which have been provided with the same statistical properties (i.e. mean value and standard deviation) and length as the original temperature series (see also Höppner and Bittner, 2007). For every period, the 99 % quantile of the wavelet intensities in the random spectra is considered as the level of significance. For a time series of 100 min, for example, this means that the 99 % quantile is calculated based on 1100 values for every period.

The mean nocturnal value of the gravity wave activity in the period range $[\tau_1; \tau_2]$ is retrieved by calculating the averaged significant wavelet intensity between $\tau_1$ and $\tau_2$ throughout the analyzed night length. The spectra are altered by randomly varying each temperature value within its error bar (4.5 K at maximum). The mean deviation of ten altered spectra from the original spectrum is taken as a measure of the uncertainty for the mean nocturnal value. Later in this publication we calculate monthly mean values. Here we use the standard error of the mean ($\sigma/\sqrt{N}$; with $\sigma$ being the standard deviation and $N$ the number of values), which is larger than the uncertainty resulting from the individual error bars.

The short-period limit of gravity waves is defined by the Brunt-Väisälä frequency, which ranges between 4 and 5 min in the UMLT (Wüst et al., 2017b). We restrict our analysis to periods of at least 6 min. The upper limit of 240 min (4 h), which we chose for gravity wave periods in the first run, is the minimum length of the analyzed nocturnal temperature series. This upper limit is raised to 480 min (8 h) in a second analysis. The influence of tides can be tentatively neglected as we limit our analyses to periods below 8 h. Apart from that, Offermann et al. (2009) note on basis of the Global Scale Wave Model (GSWM) in combination with a climatology based on satellite data of TIMED-SABER (Thermosphere Ionosphere Mesosphere Energetics Dynamics, Sounding of the Atmosphere using Broadband Emission Radiometry) that the tidal influence is small compared to gravity wave signatures at extratropical latitudes.

The examination of the response behaviour of the wavelet analysis using synthetic test data sets revealed that oscillations with shorter periods yield slightly higher peak intensities in the wavelet spectrum than oscillations with longer periods having the same amplitude. Our tests have shown that the peak wavelet intensity decays linearly for increasing periods. This effect is strongest for short time series and weakens for longer time series. In the worst case – a time series of 240 min length – the peak intensity of a 480 min signal is 34 % of the peak intensity of a 6 min signal having the same amplitude. We attribute this to be an artefact due to boundary effects, which occurs as long as the time series is not much longer than the periods analysed. Furthermore, the response peak is blurred over a wider range of periods for longer periodicities. This

makes it difficult to link absolute values of wavelet intensity to actual temperature amplitudes of the respective oscillations. However, in this work we focus on the relative behaviour of period-resolved wave activity. Additional calculations (not shown here) have shown that the period-dependence of the wavelet response is small enough not to affect the resulting behaviour of gravity wave activity.

5   Figure 2 shows the nocturnal means of the significant wavelet intensity averaged over each month for the period range $\tau \in [6\,\text{min}; 480\,\text{min}]$ with $\Delta\tau = 1\,\text{min}$ for each station. The overall behaviour at the different observation sites is quite similar. The mean wavelet intensity is close to zero for periods shorter than 25 min and starts increasing for longer periods. While there is hardly any variability on monthly scales for gravity waves with periods below 60 min, a semi-annual cycle emerges for periods longer than 60 min, which is characterized by maximum values in winter and summer. This semi-annual

10   cycle gradually turns into an annual cycle with a strong maximum during winter and minimum values during summer for gravity wave periods longer than ca. 200 min (230 min in the case of SBO, 160 min in the case of ABA). This spectrally dependent evolution of the intra-annual shape of gravity wave activity can be well recognized when averaging the monthly values over all years. This is demonstrated in Figure 3 using OPN data.

The standard deviation of the monthly mean values of significant wavelet intensity can be calculated for each period (see

15   Figure 4). As Figure 2 already suggests, the standard deviation of the monthly mean gravity wave activity is mostly increasing for larger periods. Similar to the mean value of wavelet intensity the standard deviation begins to increase remarkably at a period of 25 min. For periods longer than 60 min, the curves start separating from one another rather than following a common course. This supports the approach of analysing gravity waves with periods shorter and longer than 60 min separately as Wüst et al. (2016, 2017a) did. Furthermore, local maxima and minima are visible in the standard

20   deviation graphs (Figure 4).

## 4 Discussion

Wüst et al. (2016, 2017a) calculated the GWPED for the same measurement nights acquired at OPN, UFS, ALR, OHP and TAV as we did. The authors applied a combination of different sliding mean filters to the temperature time series and distinguished between the short-period (shorter than 60 min) and long-period (longer than 60 min) wave range. In order to compare our results to the ones of those authors, we calculated the nocturnal mean values of wavelet intensity in the period ranges 6–60 min, 60–240 min and 240–480 min and averaged the monthly mean values over all years for all stations (Figure 5). For the stations with data coverage throughout the entire year, the wavelet intensity averaged for periods 6–60 min shows very low seasonal variation with a weak maximum in June or July (also NEU), whereas a semi-annual oscillation with maxima during winter and summer is visible when averaging wavelet intensity in the period range 60–240 min. This can even be observed for TAV despite the seasonally inhomogeneous data coverage. A dominant annual variation with a winter maximum can be recognized for ABA, OHP, OPN and UFS when averaging the wavelet intensity between periods of 240 and 480 min. For SBO and TAV this statement is difficult to confirm within the given error bars. As concerns the polar stations ALR and NEU, wavelet intensity in the period ranges 60–240 min and 240–480 min is higher during the winter months of the respective hemisphere than in spring and autumn. Due to the missing data during polar summer there is no information about a secondary summer maximum.

As concerns the direct comparison between the different observation sites, which are mostly situated in or near mountainous regions, there are hardly any systematic differences in the intra-annual cycle even though the instruments are deployed at different parts of Europe. This supports the concept that although being a rather small-scale dynamical feature itself, the overall activity of gravity waves in the UMLT is mainly shaped by large-scale mechanisms, most importantly stratospheric wind filtering. In general, orographic forcing may be perceived to be a major source of gravity waves at most stations. Such source regions would be the Alps for OPN, UFS, SBO and OHP, the Caucasus for ABA, the Scandinavian Mountains for ALR and the mountains in the north of Queen Maud Land for NEU. However, there are some minor local deviations. As one may deduce from Figure 5 at ABA the increase of the mean wavelet intensity from periods 60–240 min to periods 240–480 min in most months is a bit higher than for the other mid-latitude stations. Given the fact that the FoV at ABA is located above a position that lies between the Greater and the Lesser Caucasus orographic gravity wave forcing may be even larger than for the other stations. As concerns the stations at high latitudes - ALR and NEU - the polar vortex could additionally act as a strong source of gravity waves. At TAV orographic forcing is expected to play a minor role since the terrain is flatter and wind comes predominantly from the coast. The lack of orographic waves compared to the other stations could explain the deviation from the clear annual patterns as observed in Figure 5. However, the data base at TAV is rather small (Figure 1). Further observations will have to be awaited to validate the seasonal cycles. At this moment we cannot explain the unusual low value in September, which appears at none of the other stations.

In contrast to Wüst et al. (2016, 2017a), we need nearly contiguous time series of 240 min (nearly contiguous means that we interpolate data gaps of up to 6 min) for our analysis and the wavelet method is only applied to the longest of such episodes

of a night (see section 2). Especially during winter when the weather is cloudy, this leads to differences in the data bases used by Wüst et al. (2016, 2017a) and this work. Our data base is smaller by 33 %. However, a systematic influence of the differing data bases could be excluded by applying the GWPED algorithm of Wüst et al. (2016, 2017a) to our smaller data base: the observed seasonal cycles remained persistent (not shown). Hence, our findings agree well with the results of Wüst et al. (2016, 2017a) who hardly find any seasonal variability of short-period GWPED and a dominant annual cycle with a maximum during winter in the long-period case. Even minor features like their secondary peak in May at OHP are similar to our cycles of long-period GWPED. The agreement with the findings of Wüst et al. (2016, 2017a) is an important verification of our wavelet approach being well suited for the estimation of gravity wave activity.

There are a number of further publications supporting both, the observation of an annual and of a semi-annual oscillation of gravity wave activity. Rauthe et al. (2008) discovered an annual cycle with winter maxima in temperature variations between 35 and 90 km height when analyzing intervals of 3 – 5 h at mid-latitudes. The year before Hibbins et al. (2007) published wind variations derived from radar data, which also exhibit an annual variation with a winter maximum in the altitude range 74 – 94 km above Rothera, Antarctica. They analysed the spectral range between 4 min and 8 h. Beldon & Mitchell (2009) point out that the annual mode may tend to be found in the mid- to low-frequency range of the gravity wave spectrum. Also above Rothera, Antarctica they found a semi-annual oscillation with a second maximum during summer after having restricted their analysis to oscillations shorter than 200 min, which fits quite well with our results. The authors consider the polar night jet as a possible reason for the annual component. Offermann et al. (2009) extracted gravity wave activity by calculating the standard deviation of mesopause temperatures, which also shows a semi-annual behavior with a primary summer maximum and a secondary but still strong winter maximum. Gavrilov et al. (2004) found a semi-annual behavior of wind variances between 10 min and 5 h below 82 – 85 km altitude above Hawaii. Hoffmann et al. (2010) presented an annual variation with a secondary summer maximum in wind variances between 3 and 9 h at an altitude of 80 - 100 km above Andenes, Norway and Juliusruh, Germany, which is enhanced when only looking at periods below 2 h. They conclude from their own and from preceding work that the summer maximum of gravity wave activity seems to be dominated by waves with periods smaller than 6 h. Similar results have been reported by Manson & Meek (1993) on the basis of radar measurements: a strong semi-annual variation of wave periods 10 - 100 min with solsticial maxima has been found around 87 km height, as well as dominant winter maxima and secondary summer maxima for wave periods 2 - 6 h. A semi-annual cycle is even observed with OH airglow imagers for short periods 5 - 30 min (Nakamura et al., 1999).

All these authors agree in finding enhanced gravity wave activity during winter. This could be attributed to wind filtering. During winter, the vertical profile of the zonal wind is purely eastward so that nearly all eastward-travelling gravity waves encounter critical levels in the strong westerlies and are filtered. The entire spectrum of westward propagating gravity waves however can ascend into the UMLT without encountering filtering by the zonal background wind. The stratospheric jet is reversed to a westward direction during summer, filtering out most of the westward-oriented spectrum of gravity waves. Additionally, large parts of the eastward-propagating spectrum are filtered by the tropospheric jet unless the phase velocity is high enough. This implies a higher gravity wave activity during winter compared to summer if tropospheric gravity wave

sources are considered. The assumption that this mechanism leads to a strong winter maximum for longer periods is supported by Tsuda et al. (1994), who observed the winter maximum in the period range 2-21 h and state that gravity waves in this period range are mainly generated near the ground. However, Becker & Vadas (2020) remark that especially secondary gravity waves are important in the UMLT as they yield the strongest amplitudes and vertical mixing effects of the OH* layer during winter. They are created due to intermittent body forcing or nonlinearities induced by breaking primary gravity waves (see e.g. Vadas & Fritts, 2002; Vadas et al., 2003; Franke & Robinson, 1999). Recent observations show that secondary gravity waves are often generated in the stratosphere and propagate upward into the UMLT (Chen et al., 2013, 2016; Yamashita et al., 2009; Zhao et al., 2017) where they would be observable, e.g. with OH* spectrometers.

The fact that in our investigations summer gravity wave activity for periods between 60 and 240 min is roughly as high as winter-time activity may point to a significant contribution of gravity wave sources at higher altitudes. Tsuda et al. (1994) also observe a strong summer maximum and attribute this to short-period waves in the range 5 min-2 h, which are predominantly excited at the height of the jet stream. Due to strong temperature and wind field changes even the UMLT region itself can also act as a source of gravity waves, especially in the short-periodic range (see e.g. Didebulidze et al., 2004). Recent studies based on the CMAT (Coupled Middle Atmosphere and Thermosphere) general circulation model show that the wind filtering concept as described above leads to realistic results (Medvedev & Klaassen, 2000; England et al., 2006). However, Medvedev & Klaassen (2000) remark that during summer fast gravity waves, which are able to penetrate into the mesopause, deposit wave drag of the same order of magnitude as the total eastward drag during winter due to their large amplitudes.

It is possible that secondary gravity waves also play a major role during summer. The aforementioned observations and also modeling performed by Becker & Vadas (2018) suggest that breaking orographic gravity waves in the stratosphere cause secondary waves with phase speed in the direction of the background wind, which are able to propagate to greater heights. Following this assumption, the wind fields in the stratosphere may block most of the upward propagating waves above the tropopause during summer, however the subsequent wave breaking could excite secondary waves with westward oriented phase speeds that may ascend into the UMLT. Unfortunately, with the here-presented measurements alone we can determine neither the zonal orientation nor whether periodic signatures are due to primary or secondary waves. As explained in Becker & Vadas (2018) and Vadas & Becker (2018), secondary gravity waves can either have larger scales than the primary wave, when being created by intermittent body forcing, or smaller scales when they are the product of nonlinearities accompanying primary wave breaking. According to Vadas et al. (2018), the large scale type of secondary gravity waves exhibits quite broad spectra with horizontal wavelengths between 500 and thousands of kilometres and horizontal phase speeds between 50 and 250 m/s. This corresponds to the larger part of the period range addressed in this work (periods > 33 min). Due to the large horizontal phase speeds these wave can propagate long vertical distances (Vadas & Becker, 2018) and are likely to reach the OH* layer after being excited in the stratosphere. Becker & Vadas (2018) note that small scale secondary waves do not tend to propagate large distances in the vertical due to their low phase speeds. Thus, it is unlikely that we observe the small-scale type of secondary gravity waves unless they are excited directly below the OH* layer.

While the mean emission height of the OH airglow layer stays more or less constant throughout the year (Wüst et al., 2017b), lidar measurements have shown that the mesopause is located at about 86 km in summer and rises to about 100 km altitude during winter (Lübken and von Zahn, 1991; She et al., 1993). This provides the possibility for observed waves to grow to larger amplitudes in summer. Apart from this, seasonally varying sources for gravity waves like extratropical storm systems, which are more apparent during winter, may also contribute to enhanced gravity wave activity even in the UMLT (e.g. Kramer et al., 2015, 2016 and references therein). During summer, Senft & Gardner (1991) observed enhanced wave energy at periods shorter than 3 h in the mesopause, which they attribute to increased importance of tropospheric convection as a source mechanism.

The minima in gravity wave activity at the equinoxes could possibly be attributed to stratospheric wind reversal. During these times both eastward and westward travelling gravity waves with low phase speeds encounter critical level filtering (Hoffmann et al., 2010). Assuming a Gaussian-shaped distribution of gravity wave phase speeds centered around zero it follows that most of the waves are filtered (Beldon & Mitchell, 2009). Based on these assumptions it would not be surprising that wind filtering of this central part of the gravity wave spectrum during the equinoxes leads to minimum wave activity in the UMLT region. However, it has to be noted that these are theoretical considerations which cannot be proven by our measurements alone.

The reason why gravity wave activity below 60 min shows no substantial minima during the equinoxes may be that these are gravity waves with a high horizontal phase speed or which are mainly generated above the stratosphere. Both cases would leave them unaffected by the seasonal cycle of stratospheric winds. Furthermore, GRIPS cannot measure the whole spectrum of gravity waves with equal sensitivity: this is due to horizontal averaging over the FoV and vertical averaging over the OH* layer. As concerns the effect of the OH* layer, Wüst et al. (2016) show that GRIPS has reduced sensitivity for waves with short vertical wavelengths. The sensitivity is less than 70 % for vertical wavelengths below 15 km and waves with vertical wavelengths below 5 km cannot be measured at all (the authors assumed a Gaussian distribution of vertical OH* concentration). FoV averaging depends on the FoV size and the orientation of the wave fronts and leads to reduced sensitivity for horizontal wavelengths below ca. 200 km (Wüst et al., 2016). In the following, we estimate whether the period range of 6 min to 60 min is affected by both limitations. The intrinsic frequency of a gravity wave, the vertical wave number, the horizontal wave number and the Brunt-Väisälä frequency are linked via the dispersion relation (Fritts & Alexander, 2003, equation (30) for high-frequency gravity waves). As according to CIRA-86 (Committee on Space Research (COSPAR) International Reference Atmosphere; NCAS British Atmospheric Data Centre, 2006) our observed altitude range shows a zonal wind reversal we assume that the frequency observed from ground is similar to the intrinsic frequency. Using a rather small value of $0.02\,s^{-1}$ for the angular Brunt-Väisälä frequency (Wüst et al., 2017b), gravity waves with periods of 6 min have a vertical wavelength below 15 km for horizontal wavelengths shorter than 8 km. The vertical wavelength of gravity waves with periods of 60 min is smaller than 15 km for horizontal wavelengths below 170 km. In these cases, the waves are strongly affected by both filtering mechanisms (vertical and horizontal) and therefore highly reduced in their amplitude. GRIPS is therefore less sensitive to variations in this period range compared to the case of medium range periods

60-240 min. This result does hardly change when applying equation (32) in Fritts & Alexander (2003), which describes medium-frequency waves.

In the spectrally resolved distribution of standard deviation of the wavelet intensity several local maxima and minima are visible. These are in particular present for the data sets of OHP and UFS – the stations with the longest and best data coverage. The maxima might indicate periods at which gravity waves are particularly sensitive to wind filtering (their phase velocity varies around the stratospheric wind maximum assuming a tropospheric source, for example) or these periods are generated only from time to time (due to convective sources, for example), while the minima in opposite would represent periods for which gravity wave activity remains consistent throughout the year. Most minima are found at different periods for different locations. One may tentatively speculate that these can be traced back to persistent sources of gravity waves, which are not subject to seasonal variations and are individual characteristics of the respective geographical locations.

# 5 Summary and outlook

We apply a combination of MEM and wavelet analysis in order to calculate gravity wave activity with a high spectral resolution from UMLT temperature time series. The data we analysed were acquired with identically built spectrometers called GRIPS at the DLR site Oberpfaffenhofen (OPN), Germany, the Environmental Research Station Schneefernerhaus (UFS) on Mt. Zugspitze, Germany, Sonnblick Observatory (SBO), Austria, Observatoire de Haute-Provence (OHP), France, Abastumani Astrophysical Observatory, (ABA), Georgia, Tel Aviv (TAV), Israel, the Arctic Lidar Observatory for Middle Atmosphere Research ALOMAR (ALR), Norway, and at Neumayer-Station III (NEU) in the Antarctic. Most stations are situated at or near mountainous regions, which implies enhanced excitation of orographic gravity waves.

The intra-annual behaviour of gravity wave activity in the UMLT region is observed to be strongly dependent on the wave period. At nearly all stations that allow all-season observations we find a clear semi-annual pattern of gravity wave activity for periods longer than 60 min with maximum activity during winter and summer and minimum activity during spring and autumn. The semi-annual cycle turns into an annual cycle with a winter maximum and a summer minimum for longer periods. Our investigations reveal that the transition from semi-annual to annual behaviour occurs around a period of 200 min. There is hardly any seasonal variation for periods shorter than 60 min except for a weak maximum in June / July.

Although the different instruments are deployed at quite different locations (in the Alpine region, in the Lesser Caucasus, in the Antarctic, near the Scandinavian and Israel coastal plain), the overall findings agree very well. This suggests a general or global reason for the observed intra annual variations such as wind filtering. Assuming gravity waves originating from the ground, this would explain the winter maximum of wave activity in the UMLT. The maximum in summer leading to a semi-annual variation of gravity waves with periods between 60 and 240 min might be due to wave generation above the stratospheric jet. Secondary gravity waves could contribute to both solsticial maxima. In the case of ALR and NEU the polar vortex could also act as a source of gravity waves.

Local variations are visible in the variability of gravity wave periods: there exist gravity wave periods which vary more in activity with time than others.

The algorithm presented here has been applied operationally to observations since June 2018. If the GRIPS data of a measurement night are of sufficient quality (see section 2), they are automatically processed during the following night. The data products are being integrated into the NDMC at the moment, so that information about the current gravity wave activity will soon be provided at https://ndmc.dlr.de for all active GRIPS stations. The spectrally resolved values of gravity wave activity at the Alpine stations will also be included into the Alpine Environmental Data Analysis Center (AlpEnDAC, https://www.alpendac.eu) in order to complement Alpine climate research within the scope of the Virtual Alpine Observatory (VAO, https://www.vao.bayern.de).

## 6 Data availability

The data are archived at the WDC-RSAT (World Data Center for Remote Sensing of the Atmosphere). The GRIPS instruments are part of the Network for the Detection of Mesospheric Change, NDMC (https://ndmc.dlr.de).

## Author contribution

The conceptualisation of the project, the funding acquisition as well as the administration and supervision was done by MB and SW. The operability of the instrument was assured by CS. GD and CP took care of the maintenance at ABA and TAV, respectively. The algorithm was written and tested by AZ and RS. The data analysis and visualization was done by RS. The

10 interpretation of the results benefited from discussions between SW, MB, CS, and RS. The original draft of the manuscript was written by RS. Careful review of the draft was performed by all co-authors.

## Competing interests

The authors declare that they have no conflict of interest.

## Acknowledgement

15 This research received funding from the Bavarian State Ministry of the Environment and Consumer Protection by grant number TKP01KPB-70581 (Project VoCaS-ALP). Regarding the GRIPS measurements in the Antarctic the authors would like to thank Dr. Rolf Weller from the Alfred-Wegener-Institut (AWI) and the many technicians and scientists of the Neumeyer overwintering crews.

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

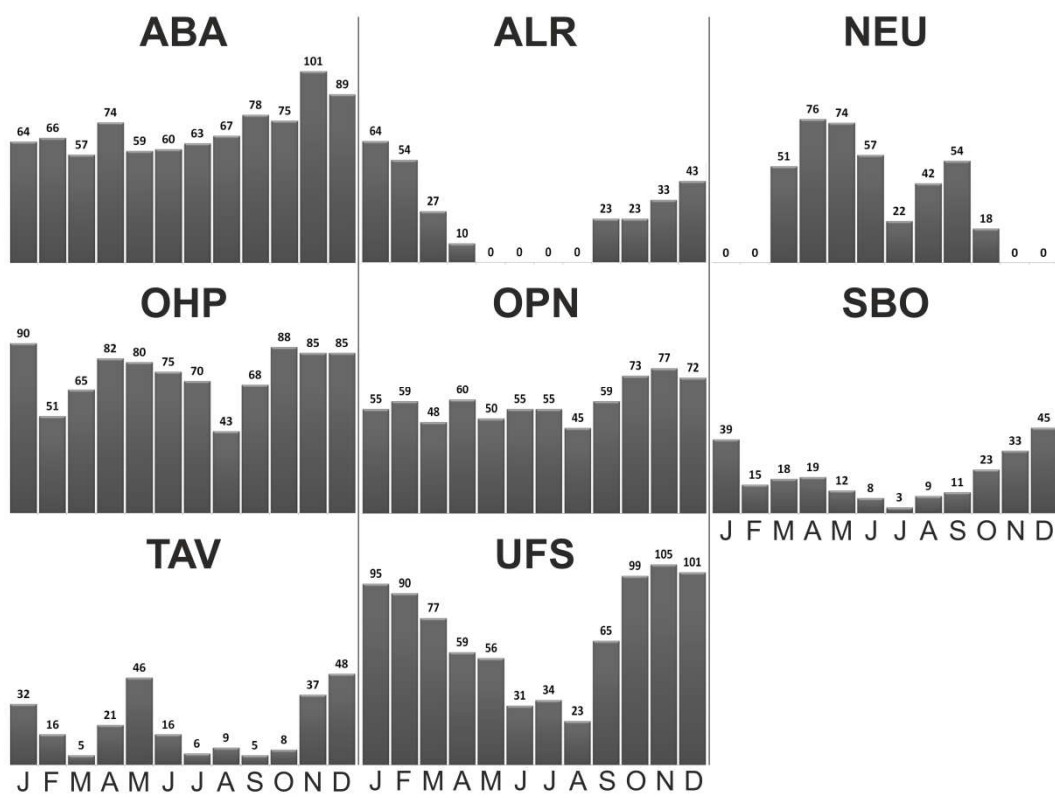

**Figure 1. Monthly data coverage (the small numbers above each column indicate the number of nights) for the individual observation sites.**

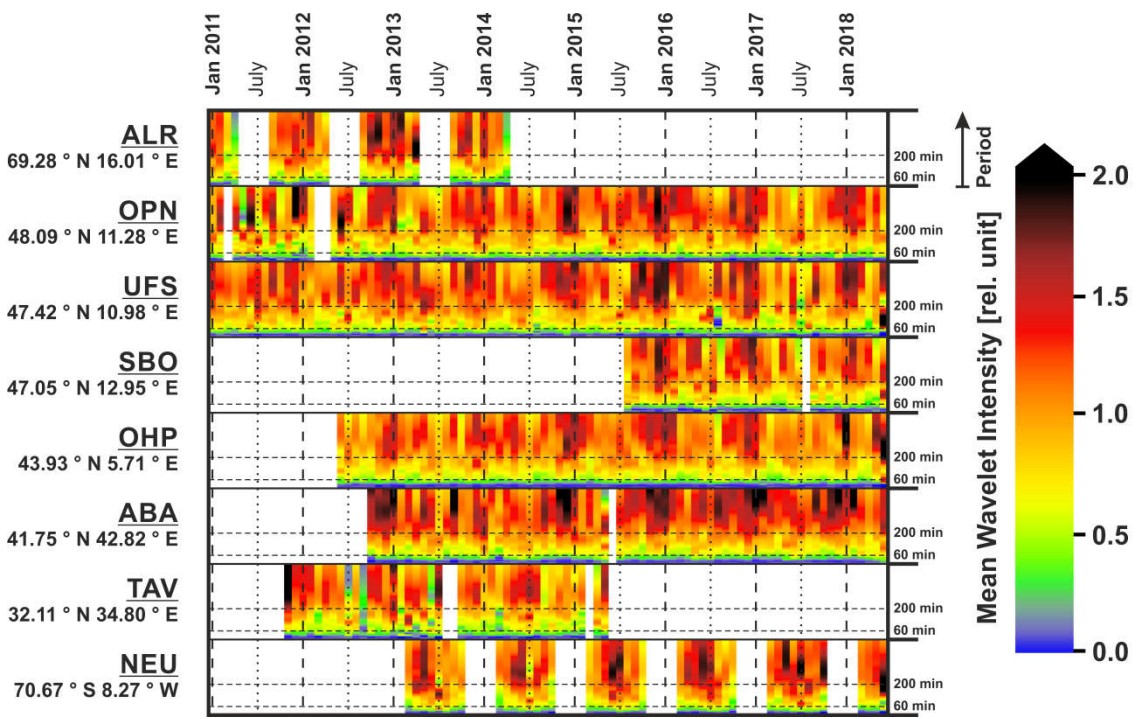

**Figure 2. Monthly mean wavelet intensity in the period range between 6 and 480 min for different observation sites.**

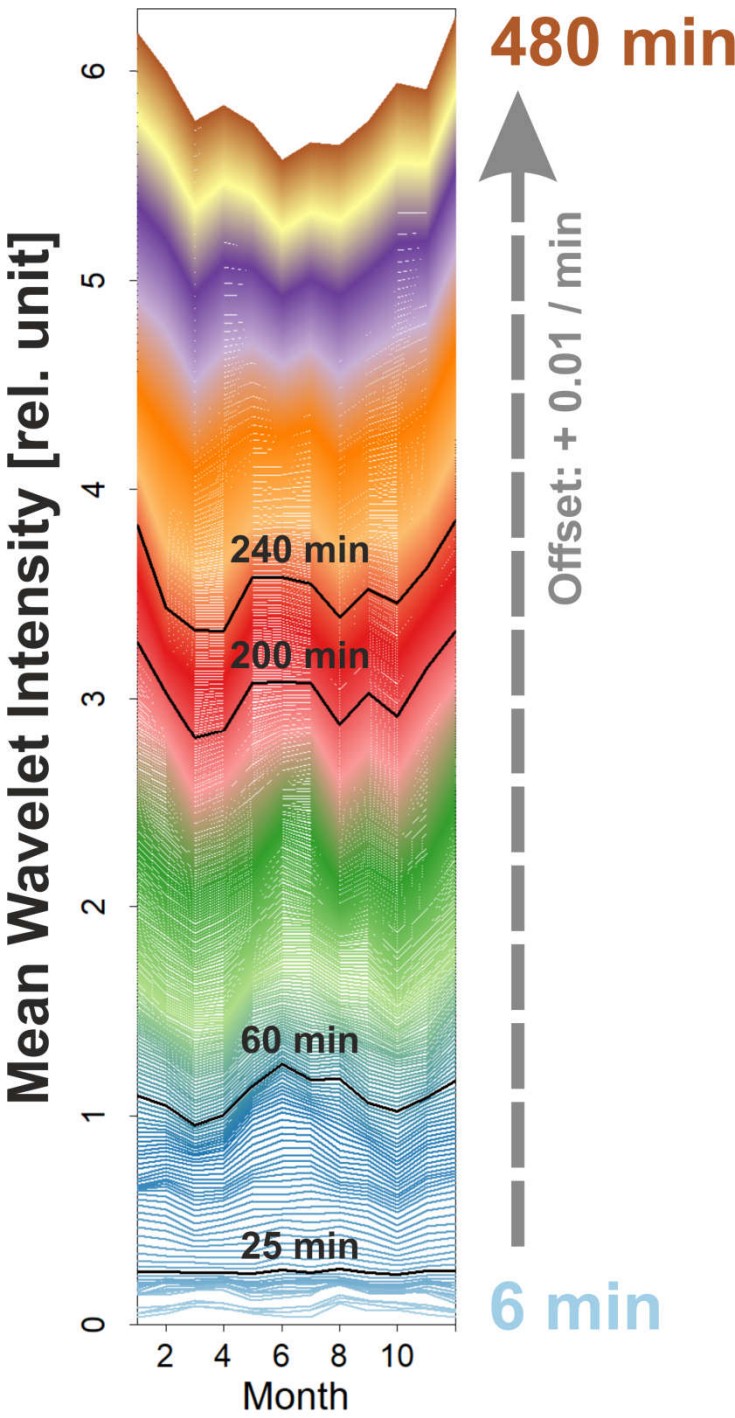

**Figure 3. Monthly mean values of nocturnal means of significant wavelet intensity, averaged over all years at OPN. The cycles are separated by gravity wave period and have been coloured and provided with an offset of 0.01 per minute of wave period to make the gradual transition of annual behaviour with growing wave periods visible.**

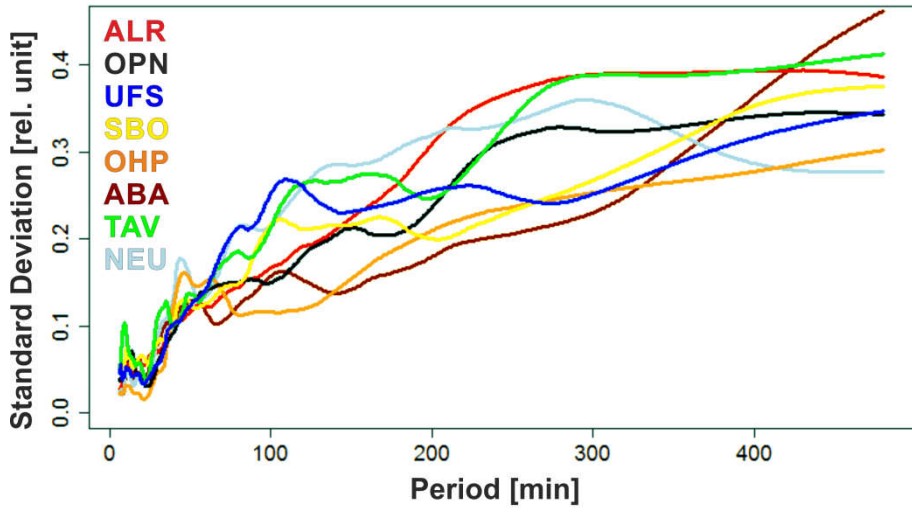

Figure 4. Standard deviation of the wavelet intensity for each period between 6 and 480 minutes and for all sites.

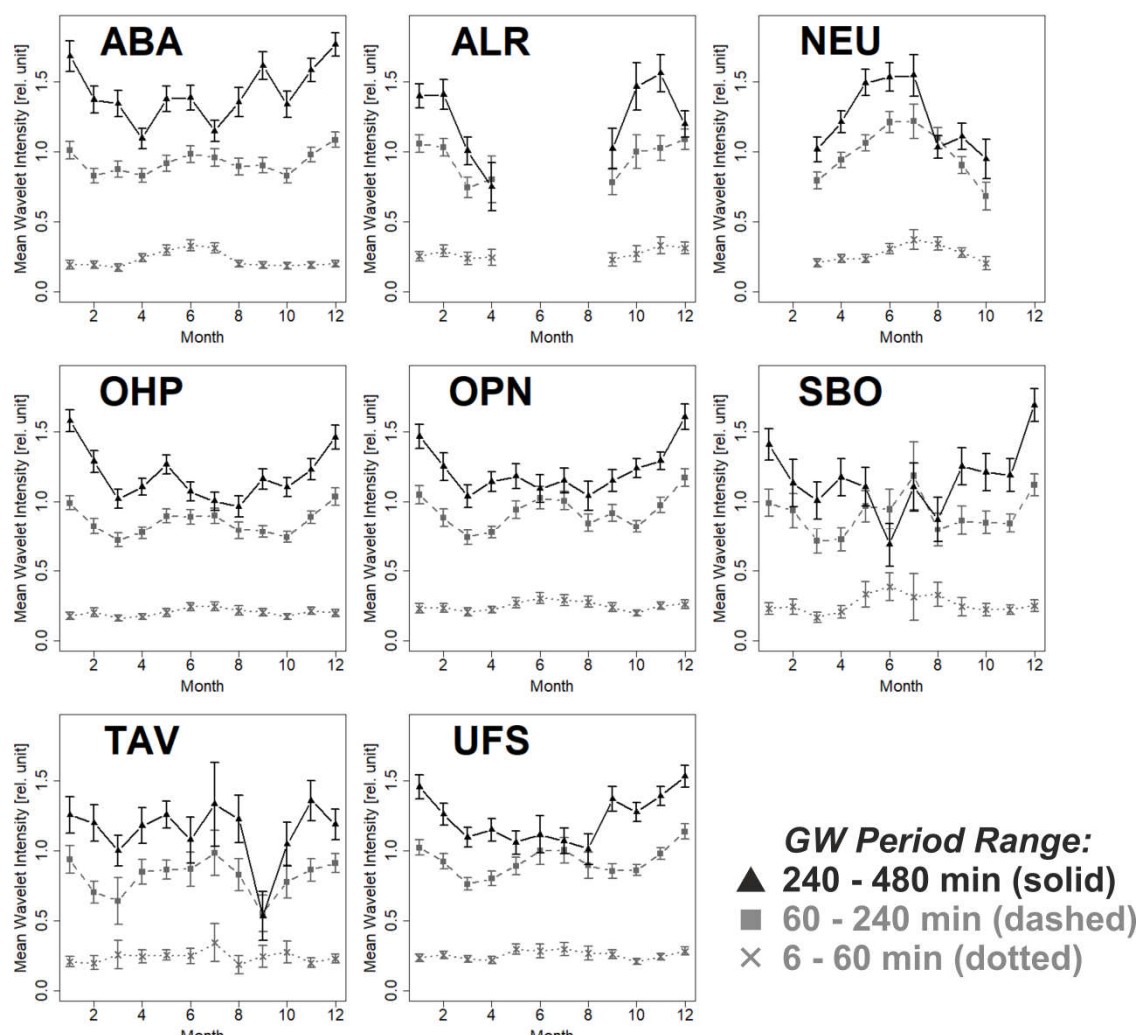

**Figure 5. Monthly mean wavelet intensity averaged in the period ranges 6–60 min, 60–240 min, and 240–480 min. The error bars represent the standard error of the mean $\sigma/\sqrt{N}$ (with $\sigma$ being the standard deviation of nocturnal values for each month and $N$ the number of nocturnal values), which is larger than the uncertainty resulting from the individual error bars of the measurements, as it is calculated in our analysis.**