# Peer review of "Intra-annual variations of spectrally resolved gravity wave activity in the UMLT region"

_Atmospheric Measurement Techniques, 2020_

## Referee Comment (RC1) · Anonymous Referee #2 · 22 Apr 2020

**Interactive comment on amt-2020-14**

General comments:

This work reports on gravity wave activity derived from hydroxyl airglow temperatures from a network of (mostly) European stations.

Seasonal variations in gravity wave activity are found to be dependent on wave period, from no seasonal variation (periods < 60min), to a semi-annual variation (periods 60-200min) and then an annual variation (periods > 200 min). This appears to be consistent across all stations and is the primary result of the work.

The paper is based on the analysis of contiguous series of OH temperature observations at these stations using a wavelet transform method, and compares the results to previous work using the same data. Although the results from each station are presented in Figs 2, 4 and 5 I think more can be discussed to compare and contrast the GW regimes between these sites.

The discussion and commentary around secondary wave generation in and above the stratosphere should be expanded. This is the key outcome of the work as one would expect the annual cycle of the stratospheric jet would lead to an annual GW activity cycle. Why should GW's in the period range 60-240min be produced above the stratospheric wind fields to produce the summer maximum in the semi-annual pattern of GW activity, but not periods below or above that range? Could there be other sources of summertime wave activity or possibly interaction between source variations, GW phase speed and the background wind? Key papers in this area by Becker and Vadas are lacking and should be referenced. Eg.

Becker, E., M. Grygalashvyly & G. R. Sonnemann, 2020: Gravity wave mixing effects on the OH*-layer. Adv. Space Res. 65, 1: 175-188, doi:10.1016/j.asr.2019.09.043.
Seems particularly relevant and should be referenced and also:

Becker, E. & S. L. Vadas, 2018: Secondary Gravity Waves in the Winter Mesosphere: Results From a High-Resolution Global Circulation Model. J. Geophys. Res. Atmos. 123, 5: 2605-2627, doi:10.1002/2017jd027460.

Vadas, S. L. & E. Becker, 2018: Numerical Modeling of the Excitation, Propagation, and Dissipation of Primary and Secondary Gravity Waves during Wintertime at McMurdo Station in the Antarctic. J. Geophys. Res. Atmos. 123: 9326-9369, doi:10.1029/2017jd027974.

Vadas, S. L. & E. Becker, 2019: Numerical Modeling of the Generation of Tertiary Gravity Waves in the Mesosphere and Thermosphere During Strong Mountain Wave Events Over the Southern Andes. J. Geophys. Res. Space Physics 124: 7687-7718, doi:10.1029/2019JA026694.

There are some references and comparisons to quite a few studies of radar derived seasonal GW variance (eg Hibbins (2007), Beldon and Mitchell (2009)) but there are many other on this topic which are relevant and could be included. For example

Tsuda, T., Murayama, Y., Nakamura, T., Vincent, R.A., Manson, A.H., Meek, C.E., Wilson, R.L., 1994. Variations of the gravity wave characteristics with height, season and latitude revealed by comparative observations. Journal of Atmospheric and Terrestrial Physics 56 (5), 555–568

Senft, D.C., Gardner, C.S., 1991. Seasonal variability of gravity wave activity and spectra in the mesopause region at Urbana. Journal of Geophysical Research 96, 17229–17264

Manson, A.H., Meek, C.E., 1993. Characteristics of gravity waves (10 min–6 h) at Saskatoon (521N, 1071W): observations by the phase coherent medium frequency radar. Journal of Geophysical Research 98 (D11), 20,357–20,367.

Thorsen, D., Franke, S.J., 1998. Climatology of mesospheric gravity wave activity over Urbana, Illinois (401N, 881W). Journal of Geophysical Research-Atmospheres 103 (D4), 3767–3780.

and also imager data for example eg.

Nakamura, T., Higashikawa, A., Matsushita, Y., 1999. Seasonal variations of gravity wave structures in OH airglow with a CCD imager at Shigaraki. Earth, Planets and Space 51, 897–906

When comparing these observations it should be considered that due to the OH layer width observations based on OH will only see waves with vertical wavelengths greater than about 14 km whereas radars will not have that selection and may see a different population of GWs.
In particular, you might be seeing low GW activity in the short period waves (< 60 mins) because they have short vertical wavelengths and the perturbation is averaged out through the OH layer.

Consider what modelling studies have been done in this area in regard to GW interaction with airglow layers and to the seasonal variability of different period bands? (GSWM has been mentioned but not its expectations for seasonal variability). The Vadas and Becker work (mentioned above) have been active in numerical modelling and are particularly relevant to the current study, and also England et al., 2006 and references therein for the CMAT model.

England, S.L., Dobbin, A., Harris, M.J., Arnold, N.F., Aylward, A.D., 2006. A study into the effects of gravity wave activity on the diurnal tide and airglow emissions in the equatorial mesosphere and lower thermosphere using the Coupled Middle Atmosphere and Thermosphere (CMAT) general circulation model. Journal of Atmospheric and Solar-Terrestrial Physics 68, 293–308

After due consideration is given to these aspects I would consider the manuscript appropriate to publish.

Further specific comments, corrections and suggestions:

Page 1. Line 28 "temporal course" -> evolution.

Page 2. Line 10. "... Region *for* gravity…"

Page 2. Line 15. The variation of wind or temperatures caused by gravity waves implies the same meaning for 'gravity wave activity'. The presence of wave activity can be detected by either parameter as both are perturbed in the same matter by the wave. The *properties* (period/speeds/wavelength) of the waves may differ subject to the sensitivity of the measuring technique to different parts of the GW spectrum.

Page 4. Line 5. The fact that measurements are only possible during the night time is not "due to solar radiation". It is due to a limitation of this technique being unable to detect the OH* emission above the solar background, and the different photochemistry involved during the daytime. This sentence needs revising.

Page 4. Heading "Data Bases"

Page 4. Line 7. Omit 'its'

Page 4. Line 19. Omit "Apart from that"

Page 4. Line 21 Omit "the"

Page 4. Line 28. How is the temperature uncertainty determined ?.

Page 4. Line 30. I don't think "succeeding" is the right word. Do you mean "successive" ?

Page 5. Line 4. "For *the* high-latitude stations (NEU,ALR)  no observations can be made during the polar summer"

Page 5 Line 5. "Also *the* Alpine stations (SBO,UFS) show *minimal* observations during summer, principally due to bad weather"

Page 6. Line 5. "wavelet analysis *method*"

Page 6. Line 6. "then delivers" -> provides or yields

Page 6. Line 16. "In the further course of " -> "Later in"

Page 6. Line 19. "short-period" omit "ic".

Page 6. Line 27+. From your synthetic data analysis you should quantify the 'slightly higher' peak intensities of shorter period waves relative to longer period waves. I.e. for a synthetic wave field of a sequence of equal amplitude waves spanning the GW spectrum region of interest what are the relative wavelet intensities for each period passing through your analysis?. That would go some way toward linking "absolute values of wavelet intensity to actual temperature amplitudes" and provide a scale for Figs 2-5 rather than just arbitrary units.

Page 7. Line 3. The term "nocturnal mean of the significant wavelet intensity" is used several times in the manuscript (eg Page 7. Line 12. and Page 8. Line 5.) but to me implies a single value (the mean of wavelet intensities that have passed the 99% significance test).

Page 7. Figure 2.
60 and 200 min period markers too small to read.
Isn't the colorbar a 'relative wave amplitude' rather than arbitrary units ?. For example do black patches signify waves that are approximately twice the amplitude as orange?.
It appears the shading has been interpolated across months which is somewhat artificial.  I would prefer to see the monthly values plotted in blocks so that month to month differences can be assessed.
Omit "Long term courses of" from the caption.

Page 7. Line 4. "average behaviour". What average ?. Do you mean the period distribution and/or seasonal variability is quite similar?.

Page 7. Line 5-6. It is very hard to distinguish anything below 60 mins, let alone 25 mins on Fig 2. Would a log-period plot show what you describe better ?.
It should be considered here what the effect of the vertical wavelength has on the observation of GW using the OH airglow layer. Short vertical wavelengths will not be seen by these observations. Are these associated with short period GW's?. Comment and discuss.

Page 7. Line 9. Dashed line at 200min for ABA is missing.

Page 7. Line 11 "using OPN data"

Figure 3. This figure does not entirely support the statements that are made in the previous paragraph. I see a strong semi-annual variation starting in the blue range (~10 mins) right through to the orange range (~350 mins), then the annual cycle above that. Indicate on the figure the 25 min, 60 min and 200 min boundaries for the change of seasonal character referred to in the text. The first sentence of the figure caption should be corrected as for Page7. Line 3.

Page 7. Line 12+. This paragraph and the selection of peaks in the standard deviations of wavelet intensity is not convincing and I question whether it is "interesting". The local peak at 45min for example appears to only occur in OHP and NEU really, and the 105 min peak only in UFS, SBO and ABA. Why are the local peaks identified of interest? Why even identify peaks in the SD of wavelet intensity which would indicate high variability and inconsistency in those periods rather than troughs which would indicate a consistent periodicity?

Page 8. Line 6. And Fig 5. Why 60-480 min and not 240-480 min range as the third interval ?. You have already covered the 60-240 min periods and you have previously identified ~200 min as the transition from semiannual to annual variation. The 60-240 and 60-480min periods are very similar as seen in fig 5.

Page 8. Line 10 course -> variation

Page 8. Line 10+ .. I would not say the annual variation is "dominant" in the stations identified, but rather that both annual and semi-annual variations are apparent (as I would expect looking at Fig 3). Perhaps if you provided the 240-480 min period range the annual cycle would be more apparent?

Page 8. Line 16 "undisturbed" -> "contiguous"

Page 8. Line 16+. I am not sure of the point of this discussion. You are using data from the same stations as Wust et al. (2016, 2017), with some further restrictions required for contiguous time series, and processing data for the same periodicities. It is no surprise that your findings agree well. If the same results and conclusions can be reached using the previous method, what are the significant advantages of the wavelet analysis method?

Page 8. Line 18. What fraction of data is unavailable for this analysis that can be analysed using the method in Wust et al. and does this highlight the limitation of this method that requires contiguous data.

Page 8. Line 24 "periodic" -> "period"

Page 8. Line 19. "us" -> "this work". Data basis -> "data base" and line 20.

Page 9. Line 12. To "a" westward

Page 9 Line 26 Minimum -> "The minima in"

Page 9 Line 27. Episodes -> times

Page 10 Paragraph 1 – as for comment Page 7. Line 12+. I did not find the "most prominent" peaks in the SD plot in Fig 4 to be particularly convincing and is it of significant interest to identify GW periods that are infrequently and inconsistently observed compared to identifying GW periods that are consistently observed (low SD) and contribute to the structure and state of the UMLT.

Page 10 Paragraph 2 – This paragraph on the comparisons between stations would be better suited following on from paragraph 1 in the discussion section (Page 8 Line 16) and expanded to compare and contrast these stations as this is the observational strength of the work. Apart from adding (or modifying) to show the 240-480 min period range, the similarities and uniqueness of each site with respect to geophysical source regions should be discussed. You attribute higher mean wavelet intensity for long-period waves to "regional peculiarities in the Caucasus" for example. The reader must assume perhaps you mean greater orographic GW forcing in this region? Why should this be the case when other sites are also located in mountainous regions? Why is there a dip in long-period GW intensity at TAV in September?. All stations appear to show a maximum in short-period (6-60 min) GW intensity in Jun-Jul, but this this does not appear to be a summer effect as the same is true for NEU. A regional map of the European sites and potential orographic GW source region may help in this discussion.  Include the polar sites of ALR and NEU in the comparison.

Page 10. Line 10. agrees with -> supports

Page 10. Line 11.  Prominently -> importantly stratospheric

Page 11. Line 3. "have been" -> were

Page 11. Line 8. The significance of being situated in mountainous regions is that these would be expected to be significant sources of orographic GW's.

Page 11. Line 9 "turns out to be" -> "is observed to be"

Page 11. Line 14. As above the 6-60min variations in Fig 5 appear to show a maximum in Jun-Jul for all stations (excluding ALR with no data in those months)

Page 11. Line 19. I do not think the distinction is made here on the basis of Figure 5.

Page 11 line 25. Your measurements are not a proxy, they are in-situ observations.

---

## Referee Comment (RC2) · Anonymous Referee #1 · 6 May 2020

The authors apply wavelet analysis to time series data of OH* rotational temperatures from eight different locations (7 northern hemisphere and 1 southern hemisphere) to extract information on the dominant periods in the data.

The periods recovered from the analysis for all eight stations have been assigned to three main categories, namely, periods less than 60 minutes, periods in the range 60-240 minutes and periods longer than 240 minutes up to the limit of the maximum data length examined (480 minutes).

Examining the pattern of gravity waves detected on an annual basis for each of the stations for the three different period classes showed no overall pattern for waves less than 25 minutes, a semi-annual cycle for waves with periods greater than 60 minutes, while waves with period greater than 200 minutes tended to show an annual cycle.

[Figure]

The uniformity of the pattern of gravity wave periods at different locations is interpreted by the authors as being likely to result from a large-scale mechanism such as atmospheric wind filtering.

The manuscript is well organised and the intention of the authors is clear. The methods used to address data gaps and to calculate the periods are clear and valid. The work is suitable for publication in AMT, provided that the specific points below are addressed.

Major Comments

The focus of the manuscript is on the classification of observed GW periods, and interpretation of the annual variation. The finding that no annual pattern was observed for period less than 25 minutes may be related to the fact that such waves may have a vertical wavelength that is less than the average depth of the OH layer, and the OH spectrometer may not be capable of detecting such waves. This may even apply to waves with periods up to 60 minutes (Page 11, line 14; "There is hardly any seasonal variation for periods shorter than 60 min."). The author should discuss this point. In doing so they might cite the results of a study by Wüst et al. (2018), and also the work of Taylor et al. (2009) and Rourke et al. (2017) who reported a relationship between the horizontal wavelength (in km) and the period (in minutes) for a range of gravity waves at several stations.

Page 8, lines 12-15; Some of the stations have a wide variation in the number of observations per month as discussed in the manuscript. How have the authors ensured that this variation does not contribute to the annual pattern recovered in the three different period categories?

Page 10, lines 8-14; the inter-comparison of the GW results from the different sites is far too short. Different possible mechanisms are mentioned earlier in the discussion (page 9, lines 8-32) but these should be considered as they apply specifically at each station in this section.

[Figure]

Page 11, line 14 (following page 9, lines 26-32); Why do the authors not observe substantial minima at the equinoxes for period less than 60 minutes?

Figures 3, 4 and 5 have y-axes with arbitrary units. This is rather unsatisfactory. Can the authors please include units that may enable others to make comparisons with their results?

Minor comments

Page 6, line 7; "resolution of 1 min in both domains". Should this not be a sampling rate of 1 min. in both domains. The Nyquist frequency is half the sampling frequency.

Page 6, line 27; please quantify and explain the origin of this effect.

Page 8, line 33; what period range was considered by Offermann et al. (2009)?

Page 9, line 3; what altitude range was considered by Hoffmann et al. (2010)?

References Rourke et al., (2017) A climatological study of short-period gravity waves and ripples at Davis Station, Antarctica (68°S, 78°E), during the (austral winter February–October) period 1999–2013. Journal of Geophysical Research: Atmospheres, 122. https://doi.org/ 10.1002/2017JD026998

Taylor et al., (2009), Characteristics of mesospheric gravity waves near the magnetic equator, Brazil, during the SpreadFEx campaign, Ann. Geophys., 27(2), 461–472, doi:10.5194/angeo-27-461-2009.

Wüst et al. (2018), Derivation of gravity wave intrinsic parameters and vertical wavelengths using a single scanning (OH(3-1) airglow spectrometer, Atmos. Meas. Tech., 11, 2937-2947, doi.org/10.5194/amt-11-2937-2018.

---

## Author Comment (AC1) · 6 Jul 2020

**Final Author's Response**

We want to thank the authors for their detailed reviews and their recommendations for improving this paper.

In this response we will first address all major points of the referees. We try to stick to the order of the comments; except for the cases both referees had the same point.

Both referees remarked that the missing seasonal dependency of gravity wave activity below 25 min and even below 60 min might also be due to these waves having vertical wavelengths shorter than the average depth of the OH\* layer and thus being not detectable with OH\* spectrometers. We also took into account FoV averaging. We took detailed account of this aspect by including the following paragraph in the discussion section.

'Furthermore, GRIPS cannot measure the whole spectrum of gravity waves with equal sensitivity: this is due to horizontal averaging over the FoV and vertical averaging over the OH\* layer. As concerns the effect of the OH\* layer, Wüst et al. (2016) show that GRIPS has reduced sensitivity for waves with short vertical wavelengths. The sensitivity is less than 70 % for vertical wavelengths below 15 km and waves with vertical wavelengths below 5 km cannot be measured at all (the authors assumed a Gaussian distribution of vertical OH\* concentration). FoV averaging depends on the FoV size and the orientation of the wave fronts and leads to reduced sensitivity for horizontal wavelengths below ca. 200 km (Wüst et al., 2016). In the following, we estimate whether the period range of 6 min to 60 min is affected by both limitations. The intrinsic frequency of a gravity wave, the vertical wave number, the horizontal wave number and the Brunt-Väisälä frequency are linked via the dispersion relation (Fritts & Alexander, 2003, equation (30) for high-frequency gravity waves). As according to CIRA-86 (Committee on Space Research (COSPAR) International Reference Atmosphere; NCAS British Atmospheric Data Centre, 2006) our observed altitude range shows a zonal wind reversal we assume that the frequency observed from ground is similar to the intrinsic frequency. Using a rather small value of 0.02 s-1 for the angular Brunt-Väisälä frequency (Wüst et al., 2017b), gravity waves with periods of 6 min have a vertical wavelength below 15 km for horizontal wavelengths shorter than 8 km. The vertical wavelength of gravity waves with periods of 60 min is smaller than 15 km for horizontal wavelengths below 170 km. In these cases, the waves are strongly affected by both filtering mechanisms (vertical and horizontal) and therefore highly reduced in their amplitude. GRIPS is therefore less sensitive to variations in this period range compared to the case of medium range periods 60 - 240 min. This result does hardly change when applying equation (32) in Fritts & Alexander (2003), which describes medium-frequency waves.'

Referee #1 asked how we ensured that the variation in the number of observations per month, that some stations exhibit, does not contribute to the seasonal patterns of gravity wave activity. For this we had a careful look at the nocturnal mean values and can confirm that e.g. winter values in the long-period range are systematically higher than the summer values so that the seasonal pattern would also occur when using only as many observations for the monthly means in winter as we observed during the summer months. The same holds for the other period ranges. This can also be seen by considering the fact that the stations with quite regular data coverage like ABA and OPN show the same seasonal behavior of gravity wave activity as the stations with a strong variation of monthly data coverage like SBO and UFS. The number of observations only contributes to the uncertainty of the monthly mean values given by  $\sigma/\sqrt{N}$ , considering mean values calculated from

less data points to more uncertain. We also ensured ourselves that the days of month are randomly distributed in our observations and show no systematic structure.

Both referees remarked that the comparison of the stations should be expanded. We elaborated this in the second paragraph of the discussion by further comparing possible gravity wave source mechanisms:

'In general, orographic forcing may be perceived to be a major source of gravity waves at most stations. Such source regions would be the Alps for OPN, UFS, SBO and OHP, the Caucasus for ABA, the Scandinavian Mountains for ALR and the mountains in the north of Queen Maud Land for NEU.'

'Given the fact that the FoV at ABA is located above a position that lies between the Greater and the Lesser Caucasus orographic gravity wave forcing may be even larger than for the other stations. As concerns the stations at high latitudes - ALR and NEU - the polar vortex could additionally act as a strong source of gravity waves. At TAV orographic forcing is expected to play a minor role since the terrain is flatter and wind comes predominantly from the coast. The lack of orographic waves compared to the other stations could explain the deviation from the clear annual patterns as observed in Figure 5. However, the data base at TAV is rather small (Figure 1). Further observations will have to be awaited to validate the seasonal cycles.'

Page 11 line 14. Referee #1 asked why we do not observe any substantial equinoctial minima for gravity wave activity in the period range 6-60 min. As a possible explanation we included

'The reason why gravity wave activity below 60 min shows no substantial minima during the equinoxes may be that these are gravity waves with a high horizontal phase speed or which are mainly generated above the stratosphere. Both cases would leave them unaffected by the seasonal cycle of stratospheric winds.'

in the discussion section on page 10. Another explication could again be short vertical wavelengths, see above.

Both referees suggested providing a physical scale to the results by linking gravity wave activity to actual temperature amplitudes. We actually tried this by applying the wavelet analysis to synthetic waves of equal amplitudes throughout the spectrum of periods, just as Referee #2 proposed. However, as we state on page 7 line 1, peaks in the wavelet spectrum exhibit a slight but period-dependent blur in the period domain. This would add further uncertainty to derived temperature amplitudes and it would be difficult to compare these temperature values to those derived by other methods. To retain consistency we decided to forego the calculation of temperature amplitudes and focus on the relative behavior of gravity wave activity.

Referee #2 liked the discussion and commentary section around secondary wave generation in and above the stratosphere to be expanded. In this part of the discussion we replaced 'altitudes above the stratospheric wind fields' by 'higher altitudes'. We put more focus on secondary gravity waves and included the proposed literature.

[revised manuscript text omitted]

**Referring to the minor comments of Referee #1:**

Page 6 line 7: The referee is right, it should rather be 'sampling rate' instead of 'resolution'. We changed this.

Page 6 line 27: We added 'This effect is strongest for short time series and weakens for longer time series. In the worst case – a time series of 240min length – the peak intensity of a 480min signal is 34% of the peak intensity of a 6min signal having the same amplitude. We attribute this to be an artefact due to boundary effects, which occurs as long as the time series is not much longer than the periods analysed.'

Page 8 line 33: Unfortunately the period range Offermann et al. (2009) addressed is hard to estimate since they do not mention it explicitly and estimate wave activity by calculating the standard deviation of temperature profiles. They rather focused on separating different wave types like planetary waves, tides and gravity waves from squared temperature standard deviations instead of resolving the exact gravity wave periods. It also has to be mentioned that Offermann et al. (2009) use vertical temperature profiles of TIMED-SABER measurements.

Page 9 line 3: The meteor radar Hoffmann et al. (2010) used addressed the altitude range 80 - 100 km. We added "at an altitude of 80 - 100 km above".

**Referring to the further specific comments of Referee #2:**

Page 1 line 28: 'temporal course' changed to 'evolution'.

Page 2 line 10: 'region of' changed to 'region for'

Page 2 line 15: We changed 'please note that the term 'gravity wave activity' does not have the same meaning in all these publications – it refers either to variations of wind or temperature caused by gravity waves' to 'please note that 'gravity wave activity' can be calculated from variations of either wind or temperature and that different measurement techniques may be sensitive to different ranges of wave parameters'.

Page 4 line 5: 'Due to solar radiation measurements are only possible during the night-time' changed to 'Measurements are only possible during the night-time since the OH\* emission would not be detectable above the solar background.'

Page 4 Heading. 'Data Basis' changed to 'Data Bases'

Page 4 line 7: omitted 'its'

Page 4 line 19: omitted 'Apart from that'

Page 4 line 21: omitted 'the'

Page 4 line 28. The temperature uncertainty is determined by Gaussian error propagation in the course of temperature retrieval as described in Schmidt et al. (2013). We added '(calculated by Gaussian error propagation)' after 'uncertainty'.

Page 4 line 30: 'succeeding' changed to 'successive'

Page 5 line 4: 'the' added

Page 5 line 5: 'Also Alpine stations (SBO, UFS) show a minimum of observations during summer. However, in these cases this effect is mainly due to bad weather.' changed to 'Also the Alpine stations (SBO, UFS) show minimal observations during summer, principally due to bad weather'

Page 6 line 5: 'method' added

Page 6 line 6: 'delivers' changed to 'provides'

Page 6 line 16: 'In the further course of' changed to 'Later in'

Page 6 line 19: 'short-periodic' changed to 'short-period'

Page 6 line 27+. Linking intensities to temperature amplitudes: see above.

Page 7 line 3. We agree that 'nocturnal mean of the significant wavelet intensity' is misleading and implies a single value. We changed 'nocturnal mean' to 'nocturnal means'.

Page 7 Figure 2. 'Long term courses of' omitted in caption. 'Arb. unit' changed to 'rel. unit' at the colorbar. '60 min' and '200 min' markers enlarged. We have changed the 2d plots to a discrete 'block-by-block' design without interpolation between the months.

Page 7 line 4: 'average' changed to 'overall'

Page 7 line 5-6: It is true that the display of very short periods is disadvantaged in this visualization. However, choosing a log plot would assign huge space to short periods < 60 min and in turn little space to medium and long periods. We decided that is would be better to properly display the period ranges with strong semi-annual and annual cycles than those with weak seasonal dependencies. Another reason is that we followed the referee's suggestion to plot the 2d spectra without interpolation and a log plot with this specification would not allow to plot the values in regular blocks next to each other. About short vertical wavelengths in the OH\* layer: see above.

Page 7 line 9. Dashed line at 200min for ABA added.

Page 7 line 11. 'using the data of OPN' changed to 'using OPN data'

Figure 3. Besides the lower and upper boundaries at 6 min and 480 min we have highlighted the periods 25 min, 60 min, 200 min and 240 min that are mentioned in the text. We omitted the original '240 min' marker. It seemed a bit confusing as it implied a linear period scale in height dimension of the image. However, the color-coded graphs for each period are separated by linear offsets in the mean wavelet intensity. Longer periods are displayed above shorter periods, but not necessarily equidistantly. Successive graphs above 25 min are less dense than e.g. above 240 min.

Page 7 Figure 3: 'arb. unit' changed to 'rel. unit' (as suggested for Figure 2). We changed the first sentence to 'Monthly mean values of nocturnal means of significant wavelet intensity, averaged over all years', hoping this formulation will be less confusing.

Page 7 line 12+. We withdrew the focus on the identification of local peaks and tried to emphasize the general behavior. As Referee #2 stated correctly, also local minima may be interesting as they are indicating periods of enhanced consistency. We changed 'It is interesting to note the occurrence of local peaks' to 'Furthermore, local maxima and minima are visible' and dropped 'A local peak of enhanced variability can be found at a period around 45 min for six out of eight stations (ABA, ALR, NEU, OHP, SBO, TAV). Another peak, or at least a strongly increased slope, is visible around a period of 105 min for four out of eight stations (ABA, SBO, TAV, UFS). Other local peaks can be found at periods of about 80 min (NEU, UFS, TAV; shoulder at ALR and SBO) and about 160 min (SBO and TAV).'.

Figure 4: As we are no longer referring to distinct periods we have removed the period markers again and omitted 'The positions of local maxima around the periods 45min, 80min, 105min and 160min are marked by dashed grey lines.' in the caption. We changed 'arb. unit' to 'rel. unit' in the y axis label.

Page 8 line 6/Figure 5. We replaced the 60-480 min range by a 240-480 min range. The caption of Fig 5 has been adapted. We changed the y axis label from 'arb. unit' to 'rel. unit'.

Further changes in this context: replaced '60-480' by '240-480' on page 8 lines 6, 12 and 13; page 10 line 16.

Page 8 line 10. 'course' changed to 'variation'.

Page 8 line 10+: see Page 8 line 6/Figure 5.

Page 8 line 16. 'undisturbed' changed to 'contiguous' (both occurrences)

Page 8 line 16+. It is true that it is not surprising that our results agree with Wüst et al. (2016, 2017a), considering our similar data sets and methods. We integrate our wavelet results over the broad period ranges and compare it to the GWPED in order to validate our method. Being sure that we get the same results, we then make use of the great advantage of the wavelet analysis and calculate gravity wave activity for resolved periods. We try to emphasize this by adding the sentence 'The agreement with the findings of Wüst et al. (2016, 2017a) is an important verification of our wavelet approach being well suited for the estimation of gravity wave activity.' at page 8 line 24.

Page 8 line 18. On average, 67 % of the nights of which potential energy density can be calculated following the method of Wüst et al. are of sufficient quality to allow the application of the heredescribed wavelet method. Thus 33 % of the data analyzed by Wüst et al. are unavailable for the wavelet method. This indeed highlights the limitation of our method. We expanded the sentence to 'Our data base is smaller by 33%.'.

Page 8 line 19: 'us' changed to 'this work'. 'data basis' changed to 'data base' in line 19 and line 20

Page 8 line 24: 'long-periodic' changed to 'long-period'

Page 9 line 12: 'a' added

Page 9 line 26: 'Minimum' changed to 'The minima in'

Page 9 line 27: 'episodes' changed to 'times'

Page 10 paragraph 1 (following page 7 line 12+): We changed 'peaks were identified' to 'maxima and minima are visible'. We dropped 'most prominently around 45 min, 80 min, 105 min, and 160 min, which occur for more than one station'. We changed 'The peaks' to 'These'. We changed 'These might be periods' to 'The maxima might indicate periods' and extended this sentence by ', while the minima in opposite would represent periods for which gravity wave activity remains consistent throughout the year. Most minima are found at different periods for different locations. One may tentatively speculate that these can be traced back to persistent sources of gravity waves, which are not subject to seasonal variations and are individual characteristics of the respective geographical locations.'.

Page 10 paragraph 2: Moved behind paragraph 1 of the discussion section. Referee #2 suggested to expand the comparison between the stations, discussing also the similarities and uniqueness of each site. We took care of this as described above. We have added the 240-480 min range to Figure 5. Specifically, Referee #2 asked, why we attribute the high intensity at ABA to orographic forcing, while also other stations are near mountainous regions. We speculate that orographic forcing at ABA may be particularly high due to the Greater Caucasus being situated north of the FoV and the Lesser Caucasus being located south of the FoV. The changes in the manuscript are described above. Referee #2 asked why there is a dip in long-period GW intensity at TAV in September. Unfortunately we have no explanation for this. We included 'At this moment we cannot explain the unusual low value in September, which appears at none of the other stations.'. Concerning the maximum during June-July in the 6-60 min period range, we changed 'summer maximum in some cases' to 'maximum in June or July (also NEU)'.

Page 10 line 10: 'agrees with' changed to 'supports'

Page 10 line 11: 'prominently' changed to 'importantly stratospheric'

Page 11 line 3: 'have been' changed to 'were'

Page 11 line 8. We expanded this sentence by ', which implies enhanced excitation of orographic gravity waves'.

Page 11 line 14: We have expanded the sentence by 'except for a weak maximum in June / July'.

Page 11 line 9: 'turns out to be' changed to 'is observed to be'

Page 11 line 14: We expanded the sentence by 'except for a weak maximum in June / July'.

Page 11 line 19: As the discussion section has been expanded we replaced 'It can explain the observed annual and semiannual modes of gravity wave activity in the UMLT under the following assumptions: gravity waves with periods between 60 and 240 min (240 min and 480 min) are generated at altitudes of or above (below) the stratospheric jet.' by 'Assuming gravity waves originating from the ground, this would explain the winter maximum of wave activity in the UMLT. The maximum in summer leading to a semi-annual variation of gravity waves with periods between 60 and 240min might be due to wave generation above the stratospheric jet. Secondary gravity waves could contribute to both solsticial maxima. In the case of ALR and NEU the polar vortex could also act as a source of gravity waves.'.

Page 11 line 25: 'proxy' changed to 'values'

---

## Author Response (AR2)

The authors are glad to hear that Anonymous Referee #2 is satisfied with our changes and thank him/her for the additional minor corrections. The marked-up version of the manuscript can be found below the comments on the minor corrections.

5   We addressed the minor corrections as follows.

Page 4 line 1: We changed 'data base(s)' to 'database(s)' at page 4 line 1, page 8 line 30, page 9 line 3, line 4 and twice in line 5.

Page 4 line 26: 'agree with' changed to 'are the same as'

Page 5 line 4: see next comment

10   Page 5 line 5: We changed the sentence 'For the high-latitudinal stations…' to the Referee's suggestion 'No observations are possible at the high-latitude stations (NEU, ABA) during the polar summer due to extended daylight hours.'

Page 6 line 21: In fact, all our analyses shown refer to what we call "second run" with periods < 480 min so that there is no need to mention the first run. We revised this paragraph by changing *'The upper limit of 240min (4h),*

15   *which we chose for gravity wave periods in the first run, is the minimum length of the analyzed nocturnal temperature series. This upper limit is raised to 480min (8h) in a second analysis. The influence of tides can be tentatively neglected as we limit our analyses to periods below 8h.'* to *'The upper period limit of 480 min (8 h) we chose is twice the minimum length of the analyzed nocturnal temperature series. The influence of tides can be tentatively neglected when limiting our analyses to periods below 8 h.'*

20   Page 6 line 23: inserted 'the'

Page 7 line 10: The Referee is right – there is still a visible semi-annual mode present for periods > 200 min. We identified 200 min as the period that marks the beginning of the gradual transition. The winter maximum (peak to seasonal minimum) is twice as large as the summer maximum at a period of 200 min, surpassing it more and more for longer periods. We refined our wording as follows:

25   - Page 1 line 25: 'occurs' changed to 'starts'
   - Page 7 line 9 ff: *'This semi-annual cycle gradually turns into an annual cycle with a strong maximum during winter and minimum values during summer for gravity wave periods longer than ca. 200 min (230 min in the case of SBO, 160 min in the case of ABA).'* changed to *'This semi-annual cycle gradually turns into an annual cycle with a strong maximum during winter and minimum values during summer*
30   *starting at a gravity wave period around ca. 200 min (230 min in the case of SBO, 160 min in the case of ABA; the difference of winter maximum and seasonal minimum is twice as large as for the summer maximum at these periods).'*
   - Page 13 line 13: 'occurs' changed to 'starts'

Page 8 line 16: 'As concerns the direct comparison … there are hardly any …' changed to 'A direct comparison
35   … shows hardly any …'

Page 8 line 24: definition added, 'FoV' changed to 'field-of-view (FoV)'

Page 9 line 14: 'may tend to be' changed to 'tends to be'

Page 10 line 13: 'short-periodic' corrected to 'short-period'

Page 10 line 24: 'here-presented measurements' changed to 'measurements presented here'

5    Page 10 line 34: 'distances in the vertical' changed to 'vertical distances'

Page 11 line 29: inserted 'the'

Page 12 line 7: 'opposite' changed to 'contrast'

[revised manuscript text omitted]
 1).  No observations are possible at the high-latitude stations (NEU, ABA) during the polar summer due to extended daylight hours. Also the Alpine stations (SBO, UFS) show minimal observations during summer, principally due to bad weather. The station at TAV exhibits a rather inhomogeneous data distribution due to stray light and technical issues (Wüst et al., 2017a).

Table 1. Start and end dates of the analysed time series for the respective station. The same start dates as in Wüst et al. (2016, 2017a) have been chosen for each data set as far as the respective stations have been analysed therein.

| Station | Instrument | Start of analysed time series | End of analysed time series | Total number of nights | Number of nights observed | Number of nights analysed |
|---------|-----------|-------------------------------|-----------------------------|-----------------------|---------------------------|---------------------------|
| ABA (41.75° N, 42.82° E) | GRIPS 5 | 2012/10/15 | 2018/06/05 | 2001 | 1974 | 853 |
| ALR (69.28° N, 16.01° E) | GRIPS 9 | 2011/01/01 | 2014/04/08 | 1192 | 875 | 277 |
| NEU (70.67° S, 8.27° W) | GRIPS 15 | 2013/03/18 | 2018/06/01 | 1900 | 1402 | 394 |
| OHP (43.93° N, 5.71° E) | GRIPS 12 | 2012/06/28 | 2018/06/01 | 2164 | 2152 | 882 |
| OPN (48.09° N, 11.28° E) | GRIPS 6 | 2011/01/08 | 2018/06/05 | 2704 | 2622 | 708 |
| SBO (47.05° N, 12.95° E) | GRIPS 16 | 2015/08/05 | 2018/05/12 | 1011 | 1006 | 235 |
| TAV (32.11° N, 34.80° E) | GRIPS 10 | 2011/11/25 | 2016/01/26 | 1523 | 1478 | 249 |
| UFS (47.42° N, 10.98° E) | GRIPS 8 | 2011/01/05 | 2018/06/02 | 2704 | 2646 | 835 |

**3 Analysis and results**

The wavelet analysis is a time-dependent spectral analysis method. In contrast to other analyses, e.g. the harmonic analysis, which assumes stationary periodic signatures (Bittner et al., 1994), the wavelet analysis can identify transient wave signals, which makes it well suited for the identification of gravity wave signatures. A comprehensive mathematical description of

5  the wavelet analysis can be found in Chui (1992). We use the wavelet analysis as it was described by Ochadlick et al. (1993) based on a Morlet wavelet and apply it to the temperature time series of each night. The wavelet analysis method then provides a two-dimensional wave spectrum that depends on time and period (sampling rate of 1 min in both domains).

In order to perform a significance test, the wavelet analysis is repeated another eleven times for randomly generated data (white noise), which have been provided with the same statistical properties (i.e. mean value and standard deviation) and

10  length as the original temperature series (see also Höppner and Bittner, 2007). For every period, the 99 % quantile of the wavelet intensities in the random spectra is considered as the level of significance. For a time series of 100 min, for example, this means that the 99 % quantile is calculated based on 1100 values for every period.

The mean nocturnal value of the gravity wave activity in the period range $[\tau_1; \tau_2]$ is retrieved by calculating the averaged significant wavelet intensity between $\tau_1$ and $\tau_2$ throughout the analyzed night length. The spectra are altered by randomly

15  varying each temperature value within its error bar (4.5 K at maximum). The mean deviation of ten altered spectra from the original spectrum is taken as a measure of the uncertainty for the mean nocturnal value. Later in this publication we calculate monthly mean values. Here we use the standard error of the mean ($\sigma/\sqrt{N}$; with $\sigma$ being the standard deviation and $N$ the number of values), which is larger than the uncertainty resulting from the individual error bars.

The short-period limit of gravity waves is defined by the Brunt-Väisälä frequency, which ranges between 4 and 5 min in the

20  UMLT (Wüst et al., 2017b). We restrict our analysis to periods of at least 6 min. The upper period limit of 480 min (4 h) we chose is twice the minimum length of the analyzed nocturnal temperature series.  The influence of tides can be tentatively neglected when limit our analyses to periods below 8 h. Apart from that, Offermann et al. (2009) note on the basis of the Global Scale Wave Model (GSWM) in combination with a climatology based on satellite data of TIMED-SABER

25  (Thermosphere Ionosphere Mesosphere Energetics Dynamics, Sounding of the Atmosphere using Broadband Emission Radiometry) that the tidal influence is small compared to gravity wave signatures at extratropical latitudes.

The examination of the response behaviour of the wavelet analysis using synthetic test data sets revealed that oscillations with shorter periods yield slightly higher peak intensities in the wavelet spectrum than oscillations with longer periods having the same amplitude. Our tests have shown that the peak wavelet intensity decays linearly for increasing periods. This

30  effect is strongest for short time series and weakens for longer time series. In the worst case – a time series of 240 min length – the peak intensity of a 480 min signal is 34 % of the peak intensity of a 6 min signal having the same amplitude. We attribute this to be an artefact due to boundary effects, which occurs as long as the time series is not much longer than the periods analysed. Furthermore, the response peak is blurred over a wider range of periods for longer periodicities. This

makes it difficult to link absolute values of wavelet intensity to actual temperature amplitudes of the respective oscillations. However, in this work we focus on the relative behaviour of period-resolved wave activity. Additional calculations (not shown here) have shown that the period-dependence of the wavelet response is small enough not to affect the resulting behaviour of gravity wave activity.

5 Figure 2 shows the nocturnal means of the significant wavelet intensity averaged over each month for the period range $\tau \in [6\,\mathrm{min}; 480\,\mathrm{min}]$ with $\Delta\tau = 1\,\mathrm{min}$ for each station. The overall behaviour at the different observation sites is quite similar. The mean wavelet intensity is close to zero for periods shorter than 25 min and starts increasing for longer periods. While there is hardly any variability on monthly scales for gravity waves with periods below 60 min, a semi-annual cycle emerges for periods longer than 60 min, which is characterized by maximum values in winter and summer. This semi-annual

10 cycle gradually turns into an annual cycle with a strong maximum during winter and minimum values during summer  starting at a gravity wave periodaround ca. 200 min (230 min in the case of SBO, 160 min in the case of ABA; the difference of winter maximum and seasonal minimum is twice as large as for the summer maximum at these periods). 
[revised manuscript text omitted]